# Subcortical electrophysiological activity is detectable with high-density EEG source imaging

Martin Seeber [1], Lucia-Manuela Cantonas[1], Mauritius Hoevels[2], Thibaut Sesia[2], Veerle Visser-Vandewalle[2] & Christoph M. Michel[1,3]

Subcortical neuronal activity is highly relevant for mediating communication in large-scale brain networks. While electroencephalographic (EEG) recordings provide appropriate temporal resolution and coverage to study whole brain dynamics, the feasibility to detect subcortical signals is a matter of debate. Here, we investigate if scalp EEG can detect and correctly localize signals recorded with intracranial electrodes placed in the centromedial thalamus, and in the nucleus accumbens. Externalization of deep brain stimulation (DBS) electrodes, placed in these regions, provides the unique opportunity to record subcortical activity simultaneously with high-density (256 channel) scalp EEG. In three patients during rest with eyes closed, we found significant correlation between alpha envelopes derived from intracranial and EEG source reconstructed signals. Highest correlation was found for source signals in close proximity to the actual recording sites, given by the DBS electrode locations. Therefore, we present direct evidence that scalp EEG indeed can sense subcortical signals.

[1] Functional Brain Mapping Laboratory, Department of Fundamental Neurosciences, Campus Biotech, University of Geneva, 1201 Geneva, Switzerland. [2] Department of Stereotactic and Functional Neurosurgery, University of Cologne, 50937 Cologne, Germany. [3] Center for Biomedical Imaging (CIBM), Lausanne and Geneva, 1015 Lausanne, Switzerland. Correspondence and requests for materials should be addressed to C.M.M. (email: Christoph.Michel@unige.ch)

High-density electroencephalographic (EEG) and magnetoencephalography (MEG) systems provide us with a powerful tool to study neuronal brain dynamics with high temporal and good spatial resolution[1,2]. By applying source analysis methods to scalp recordings, neuronal activities in specific brain areas can be reconstructed with milliseconds resolution enabling real-time imaging of the temporal dynamics of whole-brain neuronal networks[3,4].

The accuracy and precision of the localization of neuronal activity using non-invasive methods is still a matter of debate. One of the gold-standards in evaluating source localization precision are intracranial recordings in patients in whom subcortical or intracranial electrodes are implanted for clinical purposes. Most frequently, that is the case in pharmacoresistant epileptic patients where intracranial recordings are used to determine the localization of the epileptic focus in the framework of pre-surgical evaluation. Such recordings give the unique possibility to compare the estimated location of the epileptic focus (in most cases the irritative zone) by scalp EEG source localization with the location of the intracranial electrodes recording the same activity with high spatial precision. While recordings are usually not performed simultaneously, the fact that the epileptic activity is supposed to be generated in a unique area allows comparison of recordings done at different time points. Such studies showed that the precision of the epileptic focus localization with high-density scalp EEG is on average ±15 mm[5]. Studies using the resected zone after successful epilepsy surgery as ground truth for localization showed that high-density EEG source imaging (ESI) correctly identified this zone with around 85% accuracy[6]. Other studies comparing EEG or MEG source localization with intracranial evoked potentials, electrocortical stimulation or functional magnetic resonance imaging (fMRI) also demonstrated impressive source localization precision using these non-invasive techniques[7–9].

The pertinent question, however, is whether activity in deeper structures of the brain can be sensed with scalp EEG or MEG measures. While it is generally believed that activity of deep brain structures is not visible from scalp recordings, several EEG and MEG studies have been published that claim to be able to determine activities in subcortical structures[10–12]. However, direct proof or disproof of this claim has not been provided yet.

In this work, we aim to investigate this question by performing simultaneous recordings from electrodes placed in subcortical regions and high-density (256 channel) scalp electrodes. Recordings were performed in Gilles de Tourette Syndrome (GTS) and in obsessive–compulsive disorder (OCD) patients undergoing electrode implantation in the thalamus and the nucleus accumbens, respectively, in the framework of deep brain stimulation (DBS) therapy. These recordings provide the unique opportunity to compare the direct local field potential recordings from these deep structures with the estimated activity of virtual electrodes in these brain areas, reconstructed from scalp EEG. In this respect, we report significant correlation between intracranial recorded and source estimated signals at solution points in proximity to the actual electrode position. Therefore, we provide direct evidence that scalp EEG indeed detects subcortical neuronal activity which can be reconstructed and located using source imaging techniques.

## Results

**Alpha oscillations are present in intracranial and scalp recordings**. Two patients suffering from GTS and two patients suffering from OCD were recorded from subcortical electrodes stereotactically targeted bilaterally in the centromedial thalamus (GTS) or in the nucleus accumbens (OCD), respectively, for deep

brain stimulation. Concerning the thalamus implantation, the planning is as such that the tip of the electrode is located at the anteromedial border of the centromedian nucleus, with the larger part of the lead being located within the Ventro-oralis internus. Concerning the nucleus accumbens implantation, the electrode is implanted at the transition of the nucleus accumbens and the internal capsule. During two or 3 days post-implantation subcortical electrodes remained externalized which provided us the opportunity for simultaneous recordings with high-density EEG during patients being at rest for 5 min with eyes closed. Intracranial electrodes were recorded in reference to the right mastoid scalp electrode and were offline recalculated to bipolar recordings. High-density scalp EEG was recorded in reference to the vertex electrode and was offline recalculated to the average reference.

We reconstructed EEG source dynamics using distributed source modeling[1,3,4,13] based on realistic head models[14,15] derived from individual MRI scans. To compare EEG source imaging results with the actual DBS electrode positions, identified from post-op computer tomography (CT), we spatially aligned imaging data from these different modalities (Fig. 1a, Supplementary Figure 1).

In all recording sessions for every participant, EEG scalp recordings showed prevalent alpha activities in the frequency range of 8–10 Hz. We found identical alpha frequencies in intracranial recordings as for scalp EEG in participants that showed a well-pronounced spectral peak (Fig. 1b, c, Supplementary Figure 2), corresponding to the well-known observation of alpha oscillations recorded in thalamic nuclei at the same time as in the cortex[16,17]. In one participant, no alpha peak was visible at intracranial sites. This subject was excluded from further analysis, since a clear spectral peak is a prerequisite for investigating the detectability of neural oscillations[18].

**Subcortical activities can be reconstructed from scalp EEG**. Time series of alpha envelopes derived from intracranial electrodes were subsequently correlated with equivalent signals derived from (scalp) EEG source reconstruction at virtual points covering the whole brain's grey matter of each subject. Visual comparison of intracranial with scalp EEG alpha envelopes (Fig. 1c) illustrated some similarities between intracranial and scalp recordings. More quantitatively, alpha envelopes derived from intracranial electrodes are significantly ($p \leq 0.01$ corrected, permutation test) correlated with EEG source reconstructed signals, located to subcortical regions in close proximity to the actual electrode sites in the nucleus accumbens (OCD) and the centromedial thalamus (GTS) (Fig. 2, Supplementary Figure 3–5). Due to the limited resolution of EEG source imaging, the spatial extent ranges beyond the exact anatomical target regions. These correlation values were maximal for time lags around zero between actual and reconstructed alpha envelopes and were diminished for larger lags (Fig. 3a) indicating a close temporal relation.

Moreover, we found significant correlation between intracranial recordings in the two hemispheres. This interhemispheric alpha correlation was present between the left and right centromedial thalamus as well as between the left and right nucleus accumbens respectively (Fig. 3a). Accordingly, EEG source images showed correlation in bilateral subcortical areas. Yet, in some cases lateralization was dependent on the electrode (left/right hemisphere) which was used for correlation (Fig. 2). No significant alpha peak, ESI correlation and intracranial cross-correlation between hemispheres could be detected in GTS2 for right intracranial derivatives. Euclidian distances between the actual intracranial electrode position and ESI correlation maximum of the closest subcortical cluster in the same hemisphere

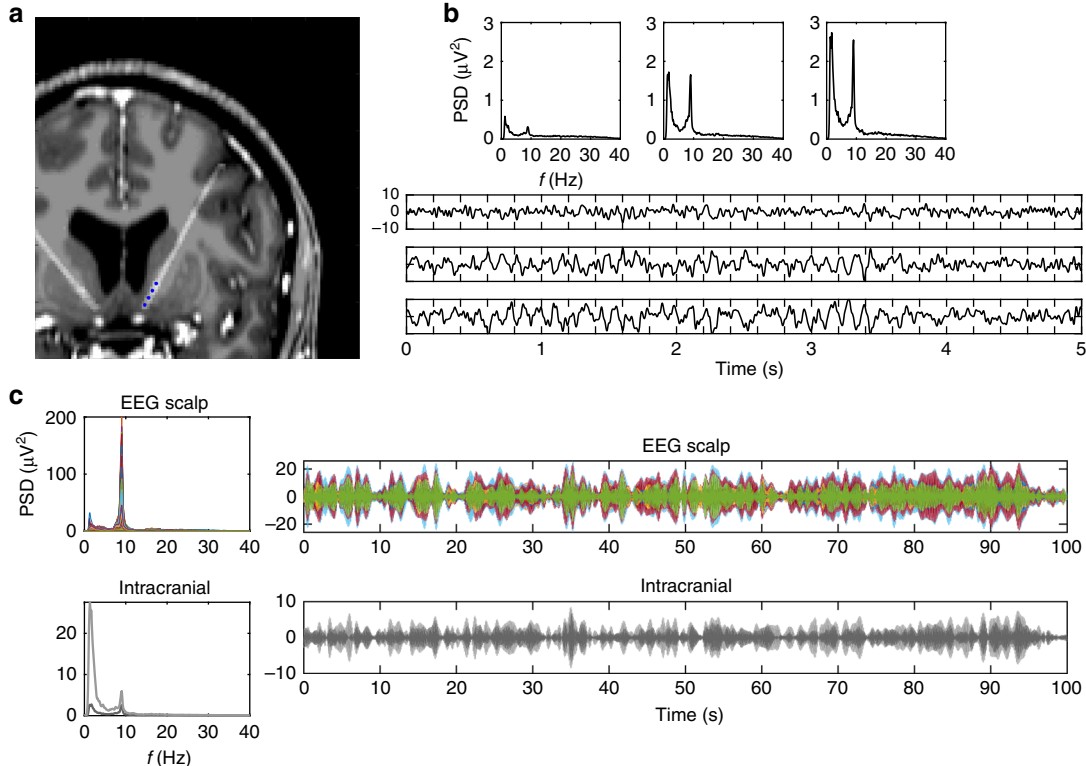

**Fig. 1** Electrode implantation and electrophysiological recordings in the OCD patient. **a** MRI overlaid with post-op CT scan, illustrating the location of the intracranial DBS electrodes indicated as blue dots, i.e., the nucleus accumbens, at its transition with the internal capsule in this case. **b** Power spectral density (PSD) and exemplary time courses are displayed from three bipolar derivations at the top, middle and lowest pair of the four intracranial electrode contacts. Note that the two most dorsal electrodes are located in the internal capsule. **c** PSD of scalp EEG and intracranial recordings were used to select individual alpha peak frequency (left panel). Exemplary time courses of alpha envelopes after narrow-band filtering showing similarities and differences of scalp and intracranial signals (right panel). Different colors are displaying different recording electrodes; light/dark gray colors correspond to left/right hemispheric implantation sites. All time courses are scaled in μV

are ranging from 14.8 to 23.5 mm and are listed in Table 1. Correlations between source reconstructed and intracranial alpha envelopes are highest in close proximity to the intracranial electrode position and decrease with increasing distance to it (Fig. 3b). Note that the steepness of this decay is indicative for the spatial resolution. Additionally, we report the spatial range in which the correlation values are significant ($p < 0.01$ corrected, permutation test) in Table 1. These ranges however are the wider, the higher the max. correlation values are.

In order to achieve robust correlation results, several seconds of recordings are needed, to capture alpha envelope dynamics which are in the range of seconds (see also Fig. 1c). While the correlation values computed from shorter time windows in the same subcortical cluster considerably vary among each other, their average correlation across windows remain stable (Fig. 3c) as well as the related source images do, being spatially highly correlated ($r > 0.9$) to each other.

## Discussion

In this study, we provide direct evidence that scalp EEG indeed can sense subcortical activities. In addition, we are able to locate these activities using EEG source imaging, if a well-pronounced signal is present at subcortical electrodes. The precision of the localization is in accordance to known uncertainties of source reconstruction techniques shown by real data[5,19] as well as with simulations for subcortical sources[10].

Interestingly, we found significant correlation between the intracranial recordings in the two hemispheres. Therefore, the bilateral localization of subcortical signals we report in the EEG source images is plausible and might stem from intrinsic dynamics emerging during minimal sensory input, since patients were instructed to close their eyes and remain as still as possible during recordings. In this state, subcortical structures are likely to default to their respective natural firing patterns. In accordance to our results, thalamic single-unit and local field potential activity were previously reported to be characterized by an oscillatory bursting pattern at low (2–7 Hz) and alpha band frequencies (8–13 Hz) recorded in the ventro oralis/centromedian parafascicular thalamic complex of TS patients under full anesthesia during surgery[20]. Further, alpha oscillations were previously reported in the human nucleus accumbens during a decision making task being coupled to gamma (40–80 Hz) amplitudes. While this alpha-gamma phase-amplitude coupling was strongest during the task, it also was reported being present in resting periods between tasks[21]. To the best of our knowledge, we provide first evidence for bilateral alpha envelop correlation in the nucleus accumbens in humans during rest.

The absence of a well-pronounced alpha peak in GTS2 at the right hemispheric electrode is concomitant with no significant cross-correlation between hemispheres and non-significant ESI correlation. This result confirms that a clear signal, i.e., alpha oscillation in this work, must be present in subcortical recordings in order to locate the dynamics of that particular signal with scalp EEG.

The finding that scalp EEG can detect subcortical sources, in our case alpha rhythms, shows that EEG contains both, cortical and subcortical signals. Previous work reported the presence of

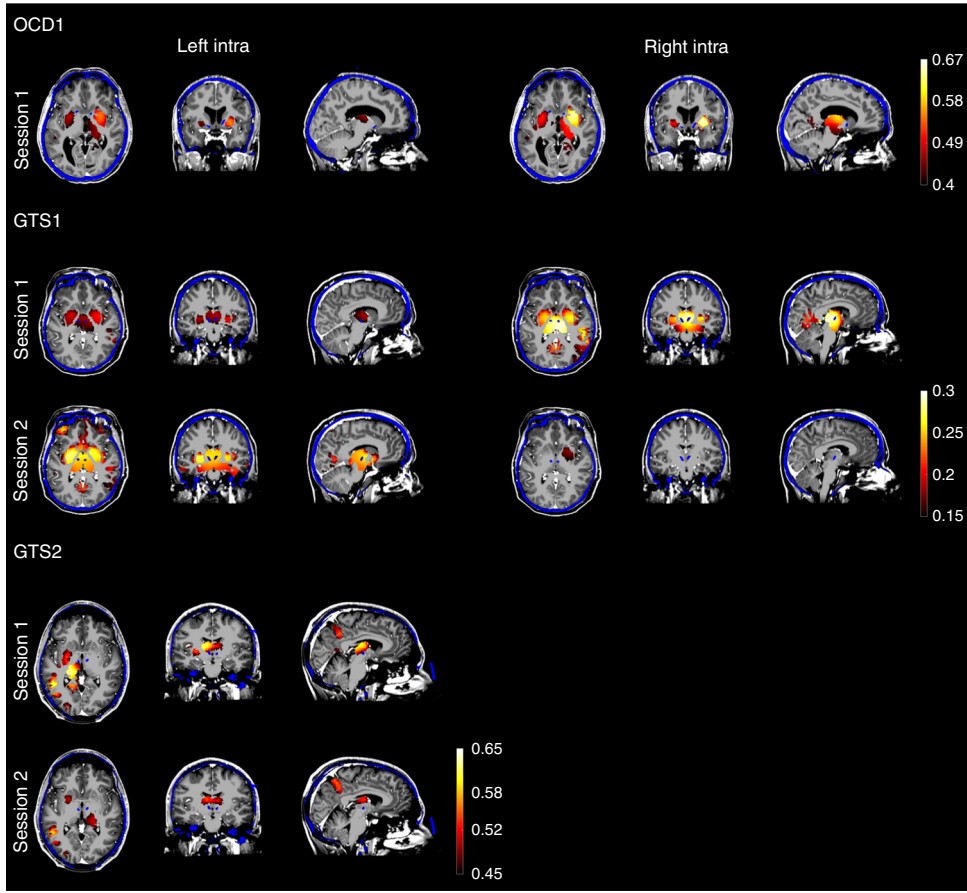

**Fig. 2** Reconstructed EEG source dynamics are correlated with actual intracranial signals. EEG source images illustrating significant correlation between source reconstructed and actually recorded alpha envelopes in subcortical areas. Intracranial electrodes are implanted in the nucleus accumbens (OCD) and in the centromedial thalamus (GTS) of the left and right hemisphere. Accordingly, highest correlations between source reconstructed and intracranial signals are located in target, i.e., implantation regions or in close proximity to it. Namely, these regions are the left/right putamen in OCD1, the left pallidum/right thalamus in GTS1, session 1, the left/right putamen in session 2 and the left thalamus in GTS2 in both recording sessions. MRI images (greyscale) are overlaid with post-op CT (blue) scans and EEG source imaging results (warm colors). Intracranial electrode positions are visible as blue dots and were used to select the view of these images

distinct cortical and thalamocortical alpha generators[16,22,23]. While our results suggest that EEG alpha waves contain thalamic activity, that does not mean that thalamocortical interactions might be the only alpha source. While volume conduction is instantaneous, neuronal transmission of thalamo-cortical interaction takes time, i.e. leading to lags. Because we used intracranial recordings as a covariate in the source reconstruction analyses to isolate thalamic signals, the EEG source images intentionally show the reconstructed counterpart of these signals, not cortical activity driven by thalamic interactions. In the seminal work by Lopes da Silva et al.[16], alpha-rhythms with the same peak frequency were recorded from both cortical and thalamic areas at the same time, but the coherences between related cortical areas were consistently larger than the thalamo-cortical coherences. This explains why our correlation analysis of intracranial and scalp EEG leads to correct identification of the subcortical sources and is not dominated by cortical alpha activity being present at the same time but loosely related to subcortical signals.

Previous work[10–12] using simulations and source reconstruction provided indirect evidence for the detectability of subcortical sources in non-invasive EEG and MEG recordings. In this work, we can confirm this finding with direct evidence from intracranial recordings providing the ground truth of subcortical signals in combination with non-invasive EEG source reconstruction locating these signals in close proximity to the actual recording sites.

## Methods

**EEG recordings**. High-density EEG was recorded using an electrode net (Geodesic Sensor Net, Electrical Geodesics Inc., Eugene, OR, USA) consisting of 256 electrodes that are interconnected by thin rubber bands and contains small sponges soaked with saline water that touch the patient's scalp surface directly.

Intracranial electrodes contained four contacts 1.5 mm and 0.5 mm apart for the OCD and GTS patients, respectively. The precise localization of the electrodes are given in Fig. 1a and Supplementary Figure 1. The cables of the subcortical electrodes exit the brain on the right parietal surface between the scalp EEG electrodes and are connected to the Physio16 box of the amplifier, while the scalp electrodes are all directly connected to the amplifier. Recordings of the EEG and physio16 box are thus synchronized and A/D converted together, leading to a precise synchronization of the signals. All data were collected at a sampling rate of 1 kHz and band-pass filtered between 0.1–100 Hz. The subcortical electrodes were all referenced to a scalp reference at the right mastoid and were offline recalculated to bipolar recordings. The scalp EEG was referenced to the vertex electrode and was offline recalculated against the average reference.

**EEG analysis**. The high-density EEG was visually inspected for artifacts. Electrodes with extensive artifacts were excluded and interpolated using a spherical spline interpolation. Afterwards, periods of artifact-free EEG were selected lasting in total at least 4 min.

We investigated one of the most prominent EEG phenomena during rest, i.e., alpha oscillations in the frequency range of 8–10 Hz, which are known to be generated in the cortex as well as in thalamic nuclei[17,24–26]. We filtered EEG and intracranial recordings in the range ±1 Hz to the individual alpha spectral peak (Fig. 1b, Supplementary Figure 2), identified with power spectral density (PSD) analyses using Welch's method[27]. Then, we computed the analytical signal using Hilbert transform and calculated the inverse solution for the real and imaginary part for each solution point in the

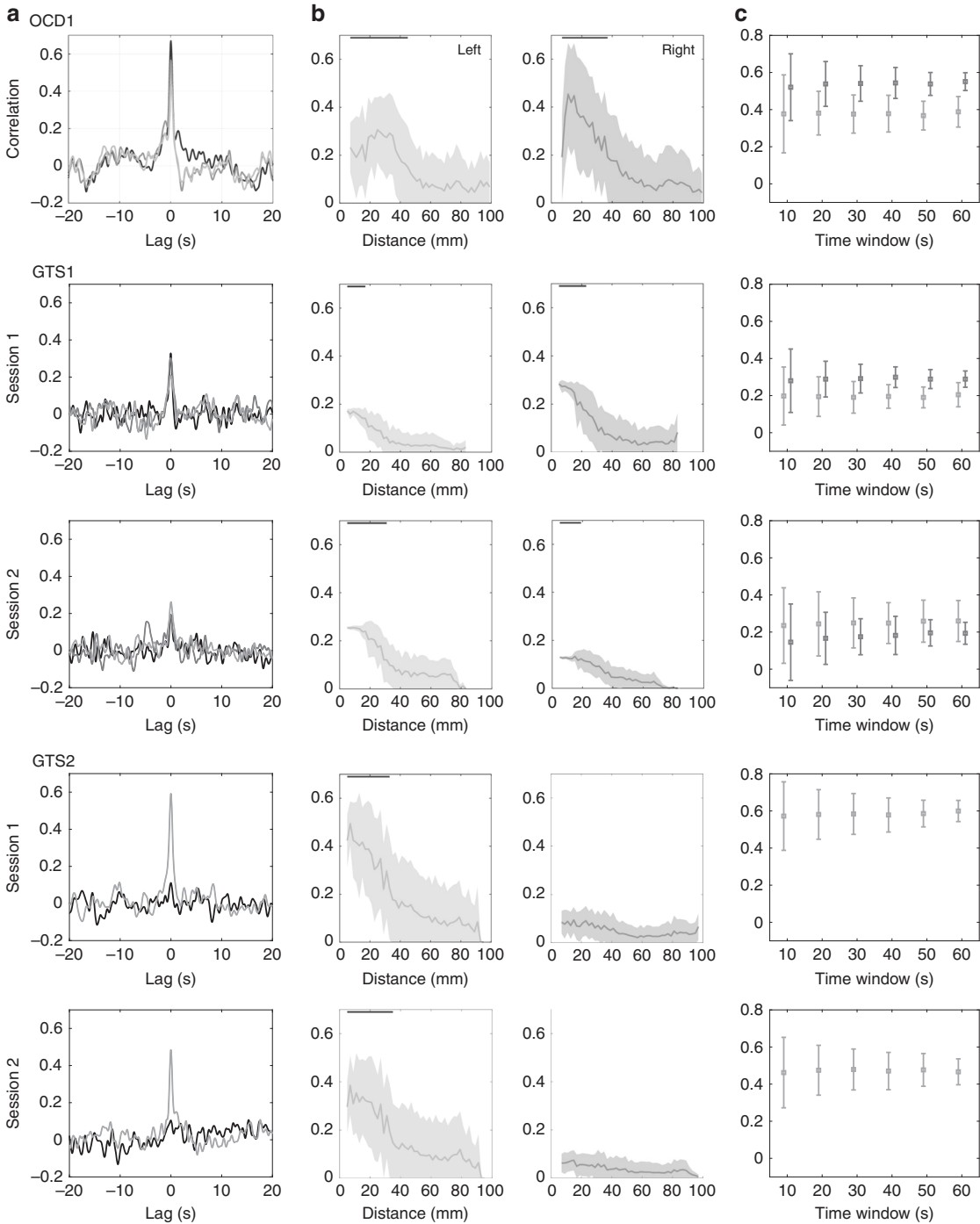

**Fig. 3** Temporal and spatial properties of the correlation analyses. **a** Cross-correlation between source reconstructed and actual alpha envelopes recorded at intracranial electrodes, placed in the nucleus accumbens (OCD) and centromedial thalamus (GTS) of the left/right hemisphere displayed in light/dark gray respectively. The cross-correlation between intracranial electrodes in the two hemispheres is illustrated in black. Note the considerable peak at lag zero representing precise temporal alignment between the correlated signals in contrast to smaller values if this alignment is disrupted at larger lags. For GTS2 no significant correlation was found in subcortical regions of the right hemisphere, neither between intracranial recordings and source reconstructed signals, nor between intracranial recordings between the two hemispheres. **b** Correlation (mean ± SD) between source reconstructed and intracranial alpha envelopes as function of the distance to intracranial electrodes. Note that the maximum in these plots represents the localization error, while the steepness of the decay with increasing distance indicates the spatial resolution in these subcortical regions. Black lines on the top show significant ranges. **c** Correlation values (mean ± SD) computed from different window sizes showing the robustness of the results with respect to different time scales

inverse space. By taking the magnitude of this complex source space signal, this leads to alpha amplitude envelopes for each solution point. The 3D (xyz) information of each solution point was combined computing the norm.

Source localization was applied using forward models based on realistic head geometry and conductivity data with consideration of skull thickness, i.e., Locally

Spherical Model with Anatomical Constraints [LSMAC][5,14,15]. Grey matter and anatomical regions were identified using volumetric segmentation of individual T1-weighted MRI ($1 \times 1 \times 1$ mm$^3$) scans, performed with the FreeSurfer image analysis suite[28–30].

The inverse solution space consisted out of about 5000 points equally distributed in this gray matter volume. The linear distributed inverse solution

**Table 1 Spatial error and range of the EEG source reconstruction**

| Subject | Left [mm] | | Right [mm] | |
|---|---|---|---|---|
| | Dist. | Range | Dist. | Range |
| OCD1 | 23 | 6–46 | 18.7 | 6–38 |
| GTS1 s1 | 14.8 | 4–18 | 23.5 | 4–24 |
| GTS1 s2 | 21.3 | 4–32 | 19.4 | 4–20 |
| GTS2 s1 | 23.5 | 4–34 | n.s. | n.s. |
| GTS2 s2 | 20.6 | 4–36 | n.s. | n.s. |

Euclidian distance in mm between intracranial electrode position and ESI correlation maximum in the closest significant subcortical cluster. Additionally, we report the spatial range around intracranial electrodes in which the correlation values are significant

LAURA[13] was used to calculate the three-dimensional (3D) current density distribution for each solution point at each moment in time.

The individual MRI (T1 and T2 weighted) with the corresponding ESI solution points were aligned with postoperative CT scans using FLIRT implemented in FSL[31,32]. Subcortical electrode coordinates were determined based on this alignment and computing a linear fit matching the 3D trajectory of the implanted lead DBS electrodes.

For intracranial recordings, we applied identical frequency analyses as for EEG scalp recordings using filtering and Hilbert transform. For each hemisphere, one bipolar deviation was selected for further analysis based on the existence of a spectral peak in the PSD, since a spectral peak is a prerequisite to demonstrate a brain oscillation within the frequency band of interest[18].

These time series of alpha envelopes derived from subcortical electrodes were subsequently correlated with their EEG source reconstructed equivalent at every solution point. To limit spatial leakage effects, we thresholded the EEG source amplitudes at their spatial average for every time frame before applying correlation with intracranial amplitude envelopes. Therefore, the resulting source images illustrate the correlation of actually recorded subcortical signals with EEG source estimated data. Negative correlation values were ignored by setting them to zero, because they might stem from the thresholding step in the analyses.

All analyses were performed using Cartool[14] and custom written MATLAB (MathWorks, Natick, Massachusetts, United States) scripts.

**Statistical analysis**. Statistical significance of correlation values were determined using nonparametric permutation tests[33,34]. In detail, we use maximum value statistics, considering the whole solution space as volume of interest and therefore correcting for multiple comparisons. To compute the permutation distribution, we shifted EEG source space and intracranial signals relative to each other in time with a randomized lag and computed correlation afterwards. Time lags ≤2 s were excluded, due to the high autocorrelation of alpha amplitude envelopes in that range given by their rather slow dynamics which are in the range of seconds (see also Fig. 1c).

Because cross-correlation at zero-lag indicates precise temporal alignment, we tested these correlation values for being significantly higher than those from random time shifts between source reconstructed and intracranial alpha envelopes, that we used to compute the permutation distribution. This procedure was repeated $10^4$ times and actual correlation values exceeding the $p = 0.01$ threshold of maximum values derived from the permutations, were considered as significant.

**Ethical approval**. The study was approved by the Ethics Committee of the University of Cologne (No. 09-155). Each patient provided informed consent to the study.

**Code availability**. EEG data analyses were performed using the freely available toolbox Cartool in combination with custom Matlab scripts, which are available from the corresponding author upon reasonable request. MRI data analyses were conducted using FreeSurfer, FSL, and MRIcron for displaying which are freely available for academic use.

**Reporting summary**. Further information on experimental design is available in the Nature Research Reporting Summary linked to this article.

## Data availability

The data that support the findings of this study are available from the corresponding author upon reasonable request.

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

## Acknowledgements

This study was supported by the Swiss National Science Foundation (Grant Number: 320030_159705) to C.M.M. and by the National Centre of Competence in Research (NCCR) "SYNAPSY–The Synaptic Basis of Mental Diseases" (NCCR Synapsy Grant # "51NF40-158776) to C.M.M.

## Author contributions

C.M.M. and V.V.V. designed research; L.C., M.S., and T.S. performed research, M.H. and M.S. analyzed data, C.M.M., M.S., T.S., and V.V.V. wrote the paper.

## Additional information

**Competing interests:** C.M.M. declares having received remuneration for a research forum organized by Philips Healthcare. The remaining authors declare no competing interests.

