## [Peer Review File · Nature Communications]

Reviewers' comments:

Reviewer #1 (Remarks to the Author):

This paper examines recent results in neural source imaging (which aim at reconstructing subcortical activity from out-of-scalp E/MEG recordings), by simultaneously recording intracranial activity and EEG (in the context of DBS therapy), and showing new evidence on highly correlated alpha activity between the DBS electrodes and reconstructed sources in their close proximity.

The paper is well-written, the results and methods are presented clearly, and the findings are quite novel. In particular, these results strengthen the position of the aforementioned body of work in source imaging, and encourage further utilization of non-invasive recordings for probing subcortical activity.

I have a few specific comments that would like the authors to address:

1) How fast would the correlations between the electrode position and other ESI sources (not necessarily the maximum) drop, as a function of distance? The drop of the correlation values as a function of the time lag between the electrode and the sources with maximum correlation are examined in Fig. 3, but it would be useful to also see the distribution of these correlations in space. One possible way to do this is to consider all ESI sources at a radius r around the electrode, and then plot the mean and STD of the correlations as a function of r (per hemisphere). This would also make the numbers reported in Table 1 more meaningful: if the correlation values drop rather rapidly around the distances reported in Table 1, one can argue that these distances can be thought of as the effective spatial resolution of subcortical source localization. But, if the drop is rather slow, Table 1 could be updated to indicate a range of distances around the electrodes, for which the correlation values are significant.

2) It would be useful to show the effect of the time windows used to compute the correlations on results of Table 1. From my reading of the methods section, the integration window for the correlations are ~4 minutes. Would the same results hold if one reduces the integration window to ~1-10 seconds? I suggest repeating the analysis for multiple scales of integration time, and assessing the robustness of the results with respect to the temporal scale.

3) Regarding lines 170-173 of the discussion: is there significant alpha activity localized near the location of the right electrode (in GTS2)? If so, the current conclusion of line 172 "This result confirms..." would not be accurate. The fact there is no significant correlation between the ESI and the nearly absent alpha activity at the right electrode does not mean that the ESI is also nearly absent. It would be useful to see if the sources near the right electrode are also significantly smaller than those near the left electrode. In that case, the conclusion would hold and would be much stronger.

4) Some minor grammatical errors/typos:

- Line 104: "More quantitative," => "More quantitatively,"
- Line 170: "GT2" => "GTS2"
- Line 454: "OCD6" => "OCD2"

Reviewer #2 (Remarks to the Author):

This Ms is very clear written and well elaborated. It addresses a question that has been a matter of debate: whether "sources" reconstructed on the basis of high resolution EEG scalp recordings (256 channels), and estimated to be localized in deep subcortical brain areas, are comparable to the local field potentials (LFPs) directly recorded in those same subcortical areas. In other words: can

subcortical LFPs by volume conduction reach the scalp EEG, and may be reconstructed based on the latter? A positive answer would give a solid basis to validate methods of high-density EEG source imaging (ESI), and validate the assumption that activity in subcortical brain areas may be estimated using EEG source imaging. The AA give sufficient information to justify a positive answer to the main question formulated above. This is an original contribution. It may have relevant implications for the interpretation of EEG source imaging methods and findings.

The AA took the opportunity to use the externalization of electrodes used for deep brain stimulation (DBS) placed in the thalamus and nucleus accumbens, to record locally electrical signals. This was done in patients who received DBS for therapeutic reasons.

The study shows that alpha envelopes derived from those intracranial electrodes (QUESTION: from which subcortical structure were the intracranial traces shown in Fig 1 recorded?; the same question applies to the traces – DBS electrodes shown in Fig 3) are significantly ($p \leq 0.01$, corrected) correlated with EEG source reconstructed signals, located to subcortical regions in close proximity to the actual electrode sites of intracranial recordings.

The name of subcortical regions should be specified. It is not enough to write DBS electrodes. Furthermore the legend of this Fig 2 should be more explicit with respect to the identification of the brain areas with their anatomical designation, where the “warm colors” are plotted. Further the legend should clearly call attention to what the reader should specifically note from these brain images, assuming that the colors correspond to the correlation values of the alpha envelopes (intracranial vs scalp EEG source reconstructed).

It should be noted that correlations of signals of this kind (this applies also to the correlation of intracranial signals recorded in the two hemispheres) may depend on the existence of a common reference. Only further along the Ms, in the section Method details, the AA explicitly write that < The subcortical electrodes were all referenced to a scalp reference at the right mastoid and were offline recalculated to bipolar recordings>. The AA should mention already in the first part of the Ms that the intracranial signals were obtained using bipolar recalculated recordings, and that the scalp EEG recordings were recalculated offline against the average reference. This information should be given from the start of the Ms since, it is essential to be able to evaluate properly the correlation results.

The AA should critically consider the fact that the cross-correlations are at zero-lag. Indeed Cross-correlations at zero-lag between EEG signals recorded from different sites can be accounted for as resulting from volume conduction. The AA write (line 73 – 76): < We expected that if the scalp EEG indeed detects subcortical neuronal activity through volume conduction, the time course of the signal at the intracranial electrode should correlate with the time course of source-estimated signals at solution points in proximity to this electrode.> Does this mean that the AA did not take into account whether time-delays different from zero might exist? In this respect it is then not clear what the AA mean with the sentence (lines 437 - 439)

<... leads to considerable cross-correlation values for small lags, if there is a significant cross-correlation present at zero lag between two signals.>

The fact that alpha rhythmic activity was recorded in the thalamus is in agreement with what is known from experimental and human studies. Regarding the Nucleus Accumbens (NA), however, this is not so simple. The study of Goto and O'Donnell (2001) to which the AA refer as showing that < (lines 167 – 169: medial thalamus and/or the hippocampus might act as pacemaker leading to synchronized activity between the left and right nucleus accumbens >, apparently implying that similar alpha-band frequency activity would be found in the NA as in the thalamic complex. Note, however, that Goto and O'Donnell's paper does not show any LFP activity at this frequency, rather just below 1 Hz and between 1 and 2 Hz (Fig 3 in that paper). Therefore it is important that the AA indicate clearly whether any of the intracranial recordings (Fig 1; Fig 3) were recorded from the NA. If this would be the case they have to give more information regarding the possibility that alpha LFPs can be generated in the NA of the awake human, and give the adequate references. To mention that the NA receives projections from thalamic nuclei is not sufficient to justify that in the NA alpha-rhythmic activity might be locally recorded.

Reviewer #3 (Remarks to the Author):

In this paper the authors correlate amplitudes of alpha rhythm estimated inside the brain from EEG

data with corresponding amplitudes measured from LFP data of intracranial recordings.

The main result is that significant results are in the vicinity of the DBS electrodes.

That, by itself, is a very interesting result regardless of the details of the methods. However, I feel that additional information is needed which should be added by the authors.

Most of all, the authors only show a few slices (those which contain the relevant DBS electrode) of significant results. The authors should show all slices. After all, it is important whether regions not in the vicinity of the LFP electrodes are also significant. It would also be helpful, if also non-significant results are shown. The corresponding figures could be included as supplementary figures.

Further details.

1. How are source directions for each voxel specified?

2. p.9 "EEG source imaging considers solely volume conduction, no active processing in cortical structures" I didn't understand that sentence. What is meant by the word "considers" here. EEG images depend on all active sources.

3. p.9 " ... is not contaminated by scalp alpha activity .. " Perhaps scalp activity is suppressed, but I doubt that there is no contamination at all.

4. "In order to exclude potential volume conduction between intracranial recordings, we applied whitening transform for signal orthogonalization." What do you mean by "whitening transform" and what has this to do with orthogonalization and volume conduction. (I am aware of the procedure by Hipp et. al., but there is no whitening.)

5. p.15 "EEG source images were demeaned in space and time, intracranial amplitude envelopes in time, before correlation." That sentence was confusing me. First, what do you mean by "images". I guess you demean the amplitudes, right? Second, in a correlation the mean is subtracted by construction. If you calculate a correlation correctly, the demeaning has no effect. Perhaps this is meant here, but what is then confusing is the phrase "demeaned in space and time". What should be done is demeaning in time for each space point. Is it this what the authors did? If not, I strongly suggest an equation here.

We thank the reviewers for their constructive comments that were tremendously helpful for improving the manuscript. In the following, we indicate our response in red font as we did for the changes in the revised manuscript.

Reviewers' comments:

Reviewer #1 (Remarks to the Author):

This paper examines recent results in neural source imaging (which aim at reconstructing subcortical activity from out-of-scalp E/MEG recordings), by simultaneously recording intracranial activity and EEG (in the context of DBS therapy), and showing new evidence on highly correlated alpha activity between the DBS electrodes and reconstructed sources in their close proximity.

The paper is well-written, the results and methods are presented clearly, and the findings are quite novel. In particular, these results strengthen the position of the aforementioned body of work in source imaging, and encourage further utilization of non-invasive recordings for probing subcortical activity.

I have a few specific comments that would like the authors to address:

1) How fast would the correlations between the electrode position and other ESI sources (not necessarily the maximum) drop, as a function of distance? The drop of the correlation values as a function of the time lag between the electrode and the sources with maximum correlation are examined in Fig. 3, but it would be useful to also see the distribution of these correlations in space. One possible way to do this is to consider all ESI sources at a radius r around the electrode, and then plot the mean and STD of the correlations as a function of r (per hemisphere). This would also make the numbers reported in Table 1 more meaningful: if the correlation values drop rather rapidly around the distances reported in Table 1, one can argue that these distances can be thought of as the effective spatial resolution of subcortical source localization. But, if the drop is rather slow, Table 1 could be updated to indicate a range of distances around the electrodes, for which the correlation values are significant.

We thank the reviewer for this comment and agree that showing the correlations as a function of the electrode distance is providing more comprehensive information. We performed these analyses for each recording, included results in Fig. 3b as mean and STD of the correlation values dependent on the distance, report the range of distances as suggested in Table 1 and added description in the result section (page 5, line 129ff). It shows that the correlation drops rather steeply with distance and are below the significance level ($p < 0.01$, corrected) at around 20-30 mm. We agree that the steepness of this drop is indicative of the spatial resolution, the range for the correlation being higher than chance however is the wider, the higher the max. correlation values are.

2) It would be useful to show the effect of the time windows used to compute the correlations on results of Table 1. From my reading of the methods section, the integration window for the correlations are ~4 minutes. Would the same results hold if one reduces the integration window to ~1-10 seconds? I suggest repeating the analysis for multiple scales of integration time, and assessing the robustness of the results with respect to the temporal scale.

As suggested by the reviewer, we repeated the analysis for multiple time scales and included these results in Fig.3c. We describe that observation in the result section (page 5, line 136ff), which reads as:

"In order to achieve robust correlation results, several seconds of recordings are needed, to capture alpha envelope dynamics which are in the range of seconds (see also Figure 1c). While the correlation values computed from shorter time windows in the same subcortical cluster considerably vary among each other, their mean correlation across windows remain stable (Figure 3c) as well as the related source images do, being spatially highly correlated ($r > 0.9$) to each other."

3) Regarding lines 170-173 of the discussion: is there significant alpha activity localized near the location of the right electrode (in GTS2)? If so, the current conclusion of line 172 "This result confirms..." would not be accurate. The fact there is no significant correlation between the ESI and the nearly absent alpha activity at the right electrode does not mean that the ESI is also nearly absent. It would be useful to see if the sources near the right electrode are also significantly smaller

than those near the left electrode. In that case, the conclusion would hold and would be much stronger.

In that matter, our initial formulation was not precise enough, so we rewrote this sentence (page 10, line 212f). We indeed did not find significant correlation between ESI alpha envelopes in right subcortical regions and the right intracranial electrode in GTS2 as written in the manuscript. We fully agree that only because we observe no significant correlation between ESI and intracranial recordings that does not mean the ESI alpha fluctuations are absent per se. However, we did not intend to claim that, but rather that if there is no clear signal already present in the intracranial recordings, in our case alpha amplitude fluctuations, we cannot identify and locate that particular signal with ESI. We did not find significant differences between right and left thalamic ESI alpha fluctuations. We interpret a spectral peak as indicative for oscillations, what we denote as “signal” in contrast to background arrhythmic activity. Not the absolute alpha power is driving the correlation but rather matching time courses between ESI reconstructed and intracranial derived alpha envelopes. We changed this sentence accordingly by highlighting i) that by “signal”, we mean alpha oscillations and ii) “the dynamics of that particular signal” which are resulting in high correlation values if ESI reconstructed and actual intracranial alpha envelope dynamics match. In addition, throughout the manuscript we made clear that, we analyzed correlations and not absolute activities.

4) Some minor grammatical errors/typos:

- Line 104: "More quantitative," => "More quantitatively,"
- Line 170: "GT2" => "GTS2"
- Line 454: "OCD6" => "OCD2"

Thank you for these corrections, we fixed these errors/typos.

Reviewer #2 (Remarks to the Author):

This Ms is very clear written and well elaborated. It addresses a question that has been a matter of debate: whether “sources” reconstructed on the basis of high resolution EEG scalp recordings (256 channels), and estimated to be localized in deep subcortical brain areas, are comparable to the local field potentials (LFPs) directly recorded in those same subcortical areas. In other words: can subcortical LFPs by volume conduction reach the scalp EEG, and may be reconstructed based on the latter? A positive answer would give a solid basis to validate methods of high-density EEG source imaging (ESI), and validate the assumption that activity in subcortical brain areas may be estimated using EEG source imaging. The AA give sufficient information to justify a positive answer to the main question formulated above. This is an original contribution. It may have relevant implications for the interpretation of EEG source imaging methods and findings.

The AA took the opportunity to use the externalization of electrodes used for deep brain stimulation (DBS) placed in the thalamus and nucleus accumbens, to record locally electrical signals. This was done in patients who received DBS for therapeutic reasons.

The study shows that alpha envelopes derived from those intracranial electrodes (QUESTION: from which subcortical structure were the intracranial traces shown in Fig 1 recorded?; the same question applies to the traces – DBS electrodes shown in Fig 3) are significantly ($p \leq 0.01$, corrected) correlated with EEG source reconstructed signals, located to subcortical regions in close proximity to the actual electrode sites of intracranial recordings.

The name of subcortical regions should be specified. It is not enough to write DBS electrodes.

Furthermore the legend of this Fig 2 should be more explicit with respect to the identification of the brain areas with their anatomical designation, where the “warm colors” are plotted. Further the legend should clearly call attention to what the reader should specifically note from these brain images, assuming that the colors correspond to the correlation values of the alpha envelopes (intracranial vs scalp EEG source reconstructed).

We agree that exact naming the subcortical structures will facilitate understanding and interpreting the mentioned figures. Following the reviewers suggestions, we named the structures, i.e. the nucleus accumbens (in OCD 1) and the centromedial nucleus of the thalamus (in GTS1 and GTS2) in the text and the figure captions of Fig.1-3. In addition, as suggested we explicitly describe and name the

target/implantation regions as well as the anatomical areas of the cluster of solution points with maximal correlation (the “warm colors”). Anatomical areas were identified using automated segmentation as we outline in the methods section on page 17, line 434. For better understanding of this figure, we also highlight more prominently that these images show alpha envelope correlations (intracranial x ESI).

It should be noted that correlations of signals of this kind (this applies also to the correlation of intracranial signals recorded in the two hemispheres) may depend on the existence of a common reference. Only further along the Ms, in the section Method details, the AA explicitly write that <The subcortical electrodes were all referenced to a scalp reference at the right mastoid and were offline recalculated to bipolar recordings>. The AA should mention already in the first part of the Ms that the intracranial signals were obtained using bipolar recalculated recordings, and that the scalp EEG recordings were recalculated offline against the average reference. This information should be given from the start of the Ms since, it is essential to be able to evaluate properly the correlation results.

We fully agree with the reviewer that the reference is essential for interpreting the results. As suggested, we include this information in the first part of the manuscript on page 3, line 90ff.

The AA should critically consider the fact that the cross-correlations are at zero-lag. Indeed Cross-correlations at zero-lag between EEG signals recorded from different sites can be accounted for as resulting from volume conduction. The AA write (line 73 – 76): < We expected that if the scalp EEG indeed detects subcortical neuronal activity through volume conduction, the time course of the signal at the intracranial electrode should correlate with the time course of source-estimated signals at solution points in proximity to this electrode.> Does this mean that the AA did not take into account whether time-delays different from zero might exist? In this respect it is then not clear what the AA mean with the sentence (lines 437 - 439)
<.... leads to considerable cross-correlation values for small lags, if there is a significant cross-correlation present at zero lag between two signals.>

We did take into account time-delay different from zero using simple correlation. In addition, we show in Fig.3a, as expected, that these correlations are maximal at zero-lag representing close temporal alignment what must be the case, since volume conduction is instantaneous. However, apparently our description was misleading. Note, that the mentioned sentence was describing how we computed the permutation distribution for statistical assessment. We rewrote that sentence in the statistics section (page 18, line 480ff) mentioning the rather slow dynamics of alpha envelopes which lead to considerable correlation values for only small lags. Meaning that the decay of the cross-correlation function (Fig.3a) is dependent of how fast the cross-correlated signals themselves do change in time.

The fact that alpha rhythmic activity was recorded in the thalamus is in agreement with what is known from experimental and human studies. Regarding the Nucleus Accumbens (NA), however, this is not so simple. The study of Goto and O'Donnell (2001) to which the AA refer as showing that < (lines 167 – 169: medial thalamus and/or the hippocampus might act as pacemaker leading to synchronized activity between the left and right nucleus accumbens >, apparently implying that similar alpha-band frequency activity would be found in the NA as in the thalamic complex. Note, however, that Goto and O'Donnell's paper does not show any LFP activity at this frequency, rather just below 1 Hz and between 1 and 2 Hz (Fig 3 in that paper). Therefore it is important that the AA indicate clearly whether any of the intracranial recordings (Fig 1; Fig 3) were recorded from the NA. If this would be the case they have to give more information regarding the possibility that alpha LFPs can be generated in the NA of the awake human, and give the adequate references. To mention that the NA receives projections from thalamic nuclei is not sufficient to justify that in the NA alpha-rhythmic activity might be locally recorded.

Indeed, we recorded alpha LFPs in the nucleus accumbens in OCD1. Following the reviewers comments we modified and expanded Fig 1, in which we now show the different contacts of intracranial electrodes and exemplary recordings as dependent of their location/implantation depth. In addition, we computed and plot power spectral densities for the different electrode contacts. In the mentioned figures, we explicitly named the anatomical target area.

Further, we removed discussion and reference to Goto and O'Donnell (2001), instead as suggested, we included discussion of previous work reporting NA alpha activity in awake humans and added corresponding reference (page 9, line 204).

Reviewer #3 (Remarks to the Author):

In this paper the authors correlate amplitudes of alpha rhythm estimated inside the brain from EEG data with corresponding amplitudes measured from LFP data of intracranial recordings. The main result is that significant results are in the vicinity of the DBS electrodes. That, by itself, is a very interesting result regardless of the details of the methods. However, I feel that additional information is needed which should be added by the authors.

Most of all, the authors only show a few slices (those which contain the relevant DBS electrode) of significant results. The authors should show all slices. After all, it is important whether regions not in the vicinity of the LFP electrodes are also significant. It would also be helpful, if also non-significant results are shown. The corresponding figures could be included as supplementary figures.

As suggested by the reviewer we include all slices, also including non-significant results in Supplementary Figure 3-5.

Further details.

1. How are source directions for each voxel specified?

We computed the norm of the 3D (xzy) signal at each voxel/solution point, since we analyzed alpha envelopes. This information is now included in the method details section (page 17, line 430).

2. p.9 "EEG source imaging considers solely volume conduction, no active processing in cortical structures" I didn't understand that sentence. What is meant by the word "considers" here. EEG images depend on all active sources.

We rewrote that sentence, since the initial formulation was indeed not optimal (page 10, line 220f).

3. p.9 "... is not contaminated by scalp alpha activity.." Perhaps scalp activity is suppressed, but I doubt that there is no contamination at all.

Probably the phrase "contamination" was not appropriate. We rephrased that sentence, being more precise in reasoning why the ESI-intracranial correlation results predominantly show subcortical, not cortical alpha dynamics, which might be driven by thalamic inputs (page 10, line 220f, 229).

4. "In order to exclude potential volume conduction between intracranial recordings, we applied whitening transform for signal orthogonalization." What do you mean by "whitening transform" and what has this to do with orthogonalization and volume conduction. (I am aware of the procedure by Hipp et. al., but there is no whitening.)

Following the comments of the reviewer, we changed this step in our analyses. Initially, we used an adapted orthogonalization method modified from Hipp et al. However, in the revised manuscript, we re-computed all results using only simple bipolar derivations, that neither were whitened, nor orthogonalized using the Hipp et al. approach. That way the analyses are i) simpler and ii) closer to the actual intracranial recordings because less processing is required. This slight methodological change only led to little changes in the ESI-intracranial correlation results, which we updated and show in Fig.1-3 and Table 1.

5. p.15 "EEG source images were demeaned in space and time, intracranial amplitude envelopes in time, before correlation." That sentence was confusing me. First, what do you mean by "images". I guess you demean the amplitudes, right?

Yes exactly, "EEG source amplitudes" is the correct term. We changed this term as suggested on page 17, line 453.

Second, in a correlation the mean is subtracted by construction. If you calculate a correlation correctly, the demeaning has no effect. Perhaps this is meant here, but what is then confusing is the phrase "demeaned in space and time". What should be done is demeaning in time for each space point. Is it this what the authors did? If not, I strongly suggest an equation here.

We agree that the initial description might not have been precise enough. Following the comment of the reviewer, we provide more detailed information and simplified our analysis for better understanding. Initially, we indeed were demeaning the EEG source amplitudes in space before correlation, which then includes demeaning in time as correctly pointed out from the reviewer.

In the revised manuscript we re-computed and replicated our results using simple thresholding instead of spatial demeaning for each time frame before correlation. Thresholding is needed in order to limit spatial smoothing/leakage effects in subcortical regions, since the spatial resolution of ESI in these regions is low. In an hypothetical extreme case, where only one signal would be present in the brain, the correlation of this signal with its EEG source reconstructed counterpart would result in exact the same correlation values at all voxels/solution points. That would be the case, only because of the smoothness of the inverse solution in combination with the normalization with the standard deviations in the correlation formula. Therefore, the more similar source reconstructed signals are *because* of spatial leakage, the blurrier they would appear in images showing correlation to intracranial recordings of that signal. By applying simple thresholding with the spatial average at each time frame, we intend to suppress 'sources' that are caused by leakage given the smoothness of the LAURA inverse solution, especially in regions distant to the electrodes, i.e. subcortical areas.

We changed the method details accordingly on page 17, line 452ff.

REVIEWERS' COMMENTS:

Reviewer #1 (Remarks to the Author):

The authors have done a great job addressing my previous comments. I have no further comments, and would like to recommend "Acceptance".

Reviewer #2 (Remarks to the Author):

The authors reviewed the Ms very appropriately. Nonetheless I have to note that the presentation of the maps of source images superimposed on MRI images and the identification of the corresponding anatomical structures would benefit from some improvement. For example, I have difficulty concluding from Fig 2 (OCD) that the " warm colors - red/yellow" dots (most clearly seen in coronal slices, both on the right and left-hand sides; in the other views the pictures are even more complex) are precisely placed in the N Accumbens. Looking in the Atlas of the Human Brain (Mai, Assheuer and Paxinos, Academic Press, 1997) I see on pages 152 -162 that the N Acc lies clearly under the tip of the frontal horn of the lateral ventricle, and just above the Olfactory Radiation and Tubercule; however, this is not what one can see looking at the "red-yellow" dots in Fig 2. It appears, rather that the " red/yellow dots" shown in Fig 2 would be situated rather at the level of the Nucleus Caudatus. /Medial Putamen above the level of the N Accumbens. In contrast, Fig 1a shows a clear localization of electrodes in the N Acc. The AA appear to acknowledge this same fact, although only partially, by writing in the caption of Fig 2 that "highest correlations between source reconstructed and intracranial signals are located in target, i.e. implantation regions or in close proximity to it. Namely, these regions are the left/right putamen in OCD1etc.". Further I also find difficult to interpret the Thalamic anatomical structures corresponding to the position of the " red/yellow" dots in GTS 1 and 2 .Of course, this does not affect the main conclusions of the Ms whether the structures where the correlations were found are exactly the N Acc or not., but it would be more accurate to name the anatomical structures in a more precise way. A suggestion is that the AA use more relative terms in the description of these structures , stating explicitly which are the target structures, but indicating, as well, that the localization of the " correlates" cannot be very precise, since these " correlates" can also involve areas in close proximity of the target area. It would also be appropriate to note that this is compatible with the data presented in Table 1, where you clear show that the Euclidian distance between an intracranial electrode and the ESI correlation maximum could extend beyond the anatomical limits of the target structure.

Reviewer #3 (Remarks to the Author):

The authors have addressed all my questions.

RESPONSE TO REVIEWERS:

Reviewer #2 (Remarks to the Author):

The authors reviewed the Ms very appropriately. Nonetheless I have to note that the presentation of the maps of source images superimposed on MRI images and the identification of the corresponding anatomical structures would benefit from some improvement. For example, I have difficulty concluding from Fig 2 (OCD) that the “ warm colors - red/yellow” dots (most clearly seen in coronal slices, both on the right and left-hand sides; in the other views the pictures are even more complex) are precisely placed in the N Accumbens. Looking in the Atlas of the Human Brain (Mai, Assheuer and Paxinos, Academic Press, 1997) I see on pages 152 -162 that the N Acc lies clearly under the tip of the frontal horn of the lateral ventricle, and just above the Olfactory Radiation and Tubercule; however, this is not what one can see looking at the “red-yellow” dots in Fig 2. It appears, rather that the “ red/yellow dots” shown in Fig 2 would be situated rather at the level of the Nucleus Caudatus. /Medial Putamen above the level of the N Accumbens. In contrast, Fig 1a shows a clear localization of electrodes in the N Acc. The AA appear to acknowledge this same fact, although only partially, by writing in the caption of Fig 2 that “highest correlations between source reconstructed and intracranial signals are located in target, i.e. implantation regions or in close proximity to it. Namely, these regions are the left/right putamen in OCD1etc.”. Further I also find difficult to interpret the Thalamic anatomical structures corresponding to the position of the “ red/yellow” dots in GTS 1 and 2 .Of course, this does not affect the main conclusions of the Ms whether the structures where the correlations were found are exactly the N Acc or not., but it would be more accurate to name the anatomical structures in a more precise way. A suggestion is that the AA use more relative terms in the description of these structures , stating explicitly which are the target structures, but indicating, as well, that the localization of the “ correlates” cannot be very precise, since these “ correlates” can also involve areas in close proximity of the target area. It would also be appropriate to note that this is compatible with the data presented in Table 1, where you clear show that the Euclidian distance between an intracranial electrode and the ESI correlation maximum could extend beyond the anatomical limits of the target structure.

We understand the reviewer's request for caution regarding the anatomical structures that are shown in Figure 2. Indeed, due to the limited spatial resolution, the regions with high correlation were not always exactly overlapping with the position of the intracranial electrodes but extended to regions in the proximity. We added a sentence in the result section explicitly stating this fact and we made sure again that nowhere in the text we claim that the maximal correlation was exactly at the place of the intracranial electrode. We already clearly stated the anatomical region in the legend of Figure 2 as the reviewer acknowledged.

The sentence added in the Result section reads:

“Due to the limited resolution of EEG source imaging, the spatial extent ranges beyond the exact anatomical target regions.”